# Face classification using electronic synapses

Peng Yao[1], Huaqiang Wu[1,2], Bin Gao[1,2], Sukru Burc Eryilmaz[3], Xueyao Huang[1], Wenqiang Zhang[1], Qingtian Zhang[1], Ning Deng[1,2], Luping Shi[2], H.-S. Philip Wong[3] & He Qian[1,2]

Conventional hardware platforms consume huge amount of energy for cognitive learning due to the data movement between the processor and the off-chip memory. Brain-inspired device technologies using analogue weight storage allow to complete cognitive tasks more efficiently. Here we present an analogue non-volatile resistive memory (an electronic synapse) with foundry friendly materials. The device shows bidirectional continuous weight modulation behaviour. Grey-scale face classification is experimentally demonstrated using an integrated 1024-cell array with parallel online training. The energy consumption within the analogue synapses for each iteration is $1,000 \times$ $(20 \times)$ lower compared to an implementation using Intel Xeon Phi processor with off-chip memory (with hypothetical on-chip digital resistive random access memory). The accuracy on test sets is close to the result using a central processing unit. These experimental results consolidate the feasibility of analogue synaptic array and pave the way toward building an energy efficient and large-scale neuromorphic system.

[1] Institute of Microelectronics, Tsinghua University, Beijing, 100084 China. [2] Center for Brain-Inspired Computing Research, Tsinghua University, Beijing 10084, China. [3] Department of Electrical Engineering and Center for Integrated Systems, Stanford University, Stanford, California 94305, USA. Correspondence and requests for materials should be addressed to H.W. (email: wuhq@tsinghua.edu.cn).

Recent advances in machine learning promise to achieve cognitive computing for a variety of intelligent tasks ranging from real-time big data analytics[1], visual recognition[2,3], to navigating the city streets for a self-driving car[4]. Currently, these demonstrations[2–5] use conventional central processing units and graphics processing units with off-chip memories to implement large-scale neural networks that are trained offline and require kilowatts of power consumption. Custom-designed neuromorphic hardware[6] with complementary metal oxide semiconductor (CMOS) technologies greatly reduces the energy consumption required. Yet, current approaches[6–10] are not scalable to the large number of synaptic weights required for solving increasingly complex problems in the coming decade[11]. The main reason that current approaches are inadequate arise from the fact that on-chip weight storage using static random access memory is area inefficient and is thus limited in memory capacity[11], and off-chip weight storage using dynamic random access memory incurs >100 times larger power consumption than on-chip memory[12]. Integrating non-volatile, analogue weight storage on-chip, in close proximity to the neuron circuits is essential for future, large-scale energy-efficient neural networks that are trained online to respond to changing input data instantly like the human brain. Meanwhile, pattern recognition tasks based on analogue resistive random access memory (RRAM) have been demonstrated either through simulations or on a small crossbar array[13,14]. However, the analogue RRAM cells still face the major challenges such as CMOS compatibility and cross-talk issues, which blocks the realization of large scale array integration. On the other hand, resistive memory arrays with relative mature technology have the problem on realizing bidirectional analogue resistance modulation[15], in which the cell conductance changes continuously in response to the SET (high conductance state to low conductance state transition) and the RESET (low conductance state to high conductance state transition) operation. This issue harms the online training function. Innovations are urgently required to find a suitable structure to combine the advantages.

In this paper, an optimized memory cell structure, which is compatible with CMOS process and has bidirectional analogue behaviour is implemented. This RRAM device[16,17] is integrated in a 1024-cell array and 960 cells are employed in a neuromorphic network[18]. The network is trained online to recognize and classify grey-scale face images from the Yale Face Database[19]. In the demonstration, we propose two programming schemes suitable for analogue resistive memory arrays: one using a write-verify method for classification performance and one without write-verify for simplifying the control system. These two programming methods are used for parallel and online weight update and both converge successfully. This network is tested with unseen face images from the database and some constructed face images with up to 31.25% noise. The accuracy is approximately equivalent to the standard computing system. Apart from the high recognition accuracy achieved, this on-chip, analogue weight storage using RRAM consumes 1,000 times less energy than an implementation of the same network using an Intel Xeon Phi processor with off-chip weight storage. The outstanding performance of this neuromorphic network mainly results from such a cell structure for reliable analogue weight storage. This bidirectional analogue RRAM array is capable of integrating with CMOS circuits to a large scale and suitable for running more complex deep neural networks[20–22].

## Results

**RRAM-based neuromorphic network.** A one-layer perceptron neural network is adopted for this hardware system demonstration, as shown in Supplementary Fig. 1. The architecture of one transistor and one resistive memory (1T1R) array, illustrated in Fig. 1a, is used to realize this neural network. The cells in a row are organized by connecting the transistor source to the source line (SL) and connecting transistor gate to the same word line (WL), while the cells in a column are organized by connecting the top electrode of the resistive memory to the bit line (BL). Figure 1a describes how the network is mapped to the 1T1R structure, that is, the input of preneuron layer, adaptable synaptic weight and weighted sum output of postneuron layer are in accordance with the pulse input from BL, cell conductance and current output through SL, respectively. Remarkable bidirectional analogue switching behaviour of our device allows us to use single 1T1R cell as a synapse to save area and energy, instead of combining two 1T1R cells as a single synapse (or weight) with differential encoding as was done in previous works[13,15]. The 1T1R array consists of 1024 cells with 128 rows and 8 columns and is optimized for bidirectional analogue switching. The arrays are constructed using fully CMOS-compatible fabrication process (see Methods section), as shown in Fig. 1b.

This network is trained to distinguish one person's face from others. The operation procedure consists of two phases: training and testing. The flow diagram of the algorithm is given in Fig. 2a. The training phase includes two subprocedures: inference and weight update. During the inference process, the nine training images (belonging to three persons) are input to the network on BL side. The activation function of the output neurons is realized

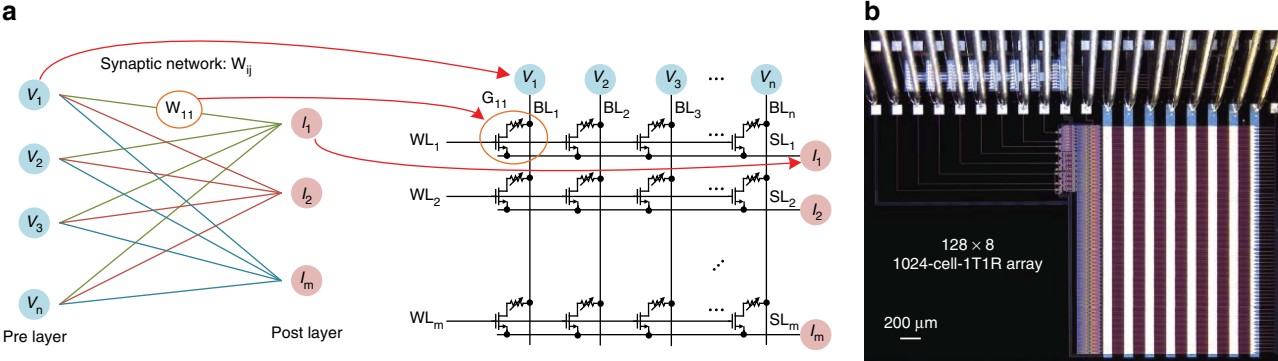

**Figure 1 | The 1T1R architecture and the 1024-cell-1T1R array.** (**a**) Mapping of a one-layer neural network on the 1T1R array, that is, the input of preneuron layer, adaptable synaptic weight and weighted sum output of postneuron layer maps to the pulse input from BL, cell conductance and current output through SL, respectively. In 1T1R, 'T' represents transistor, 'R' represents RRAM. (**b**) The micrograph of a fabricated 1024-cell-1T1R array using fully CMOS compatible fabrication process.

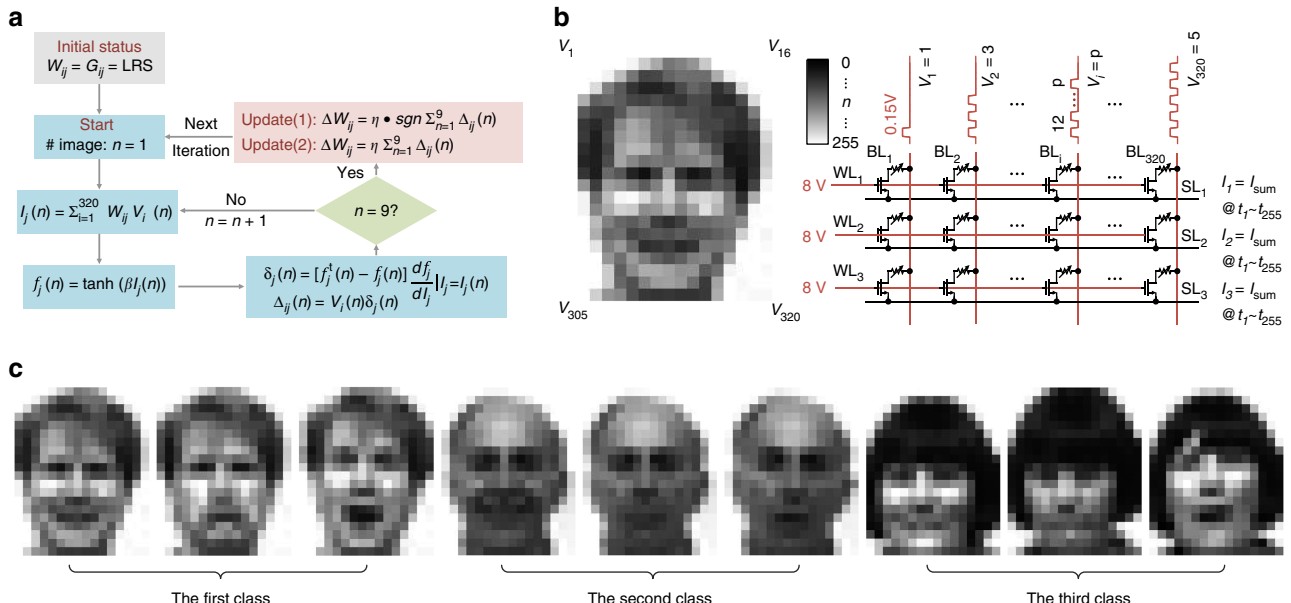

**Figure 2 | Flowchart of the perceptron model.** (**a**) The training process flow chart. In this demonstration, a batch learning model is used to accelerate the converging speed. Here 'n' represents the number of pattern, ranging from 1 to 9, 'i' implies the index of a pixel of an input pattern and can be defined from 1 to 320, 'j' is the number of output neuron that is 1–3. A correct classification during the inference phase means the active function value of a matching class of the input pattern is greater than other two classes. This network converges when all training patterns are correctly recognized. (**b**) The schematic of parallel read operation and how a pattern is mapped to the input. (**c**) The nine training images, which is a cropped and subsampled subset of the Yale Face Database[19].

by measuring the total currents on SL side (three lines) to obtain the weighted sum and applying the sum to a nonlinear activation function (tanh function) to get three output values. Each pattern is classified according to the neuron that has the largest output values. These nine images are chosen from the Yale Face Database and cropped and down-sampled to 320 pixels in $20 \times 16$ size, as Fig. 2c shows. The image is in grey scale where each pixel value ranges from 0 to 255 with smaller value corresponding to darker square. A parallel read operation (Fig. 2b) is employed for inference. The input voltage pulses are applied row by row on the fabricated array through BLs, and the total current through the SL is sensed and accumulated by a conductance linear weighting process, as the equation shows:

$$I_j(n) = \sum_{i=1}^{320} W_{ij} V_i(n). \tag{1}$$

Here $V_i(n)$ is the input signal and represents the related pixel $i$ in the pattern $n$. The pixel value leads to a matching input pulse number during the total 255 time slices to sense the weighted sum of currents, as illustrated in Fig. 2b. The total current is measured externally using the source measurement unit of a semiconductor parameter analyser, while the nonlinear activation function to the current is implemented in the software. During the weight update process, the programming of the RRAM is conducted after loading the entire nine training patterns at each iteration[23]. The programming process follows either one of the two learning rules (2) or (3) below: an update scheme using write-verify and an update scheme without write-verify.

$$\Delta W_{ij} = \eta \sum_{n=1}^{9} \Delta_{ij}(n), \tag{2}$$

$$\Delta W_{ij} = \eta \cdot \text{sgn} \sum_{n=1}^{9} \Delta_{ij}(n). \tag{3}$$

Here the learning rate $\eta$ is a constant. $\Delta_{ij}(n)$ is the calculated error between the reference output when loading the $n$th pattern and the corresponding target value determined by the pattern's label, as shown in Fig. 2a. $\Delta W_{ij}$ is the desired change for the weight connecting the neuron $i$ in input layer and the neuron $j$ in output layer. Equation (2) follows the delta rule[24] and implements both sign- and amplitude-based weight update, while equation (3) only points out the switching direction (sign-based only), following the Manhattan rule[25]. The hyper-parameters ($\beta$ controls the nonlinearity of activation function, $\eta$ is the learning rate and $f^t$ is the target value) in Fig. 2a can be found in the Methods section (test platform and the hyper-parameter values), along with the information of the platform of this demonstration.

The testing process is also a parallel read operation that reads all rows at the same time to identify the class of an input test image that is different from all the training images.

**Realization of bidirectional analogue RRAM array**. RRAM devices based on resistive switching phenomenon exhibit promising potential as the electronic synapse[26–29]. These devices have higher operation speed than the biological counterpart and they also have low energy consumption[29]. Besides, they are compatible with CMOS fabrication process[30–32] and can be scaled down[33] remarkably to reach density as high as $10^{11}$ synapses per cm$^2$. Although continuous conductance modulation behaviour on a single resistive switching device and simple neuromorphic computing on a small resistive array were reported recently[14,30], to our knowledge, large neuromorphic network utilizing the bidirectional analogue behaviour of resistive switching synapse for face classification task is not realized yet. This is due to the nature of imperfection of the device[11,13,15], such as abrupt switching during SET, the variation between each cell and the fluctuation during repeated cycles. These shortcomings have prevented the implementation of bidirectional analogue weight update and reliable update operations for a large array. Generally, the physical mechanism of the resistive switching process is attributed to the reversible modulation of the local

concentration of the oxygen vacancies in a nanoscale region[17] of the oxide. The generation or migration of a small number of oxygen vacancies in this region may induce a notable change of the conductance and thus makes the device stochastically exhibiting abrupt conductance transition step by step. This abrupt transition is more readily observed during the SET process, since the generation of each oxygen vacancy during SET process can increase the local electric field/temperature and accelerate the generation of other vacancies, and finally resulting in a large amount of oxygen vacancies formed in a short time, analogous to avalanche breakdown. This positive feedback of oxygen vacancy generation and electrical field/temperature should be effectively suppressed to avoid abrupt switching. Furthermore, the random distribution of oxygen vacancies contributes to the large variations of conductance, operation voltage and switching speed from cell to cell, which makes the system difficult to converge during the training process.

The TiN/TaO$_x$/HfAl$_y$O$_x$/TiN stacks are used as the analogue RRAM cell. All these materials are fab-friendly to enable realizing future high-density and large-scale array integration with CMOS technology. To fight against the electric field-induced avalanche breakdown during SET process, a conductive metal oxide layer is used to enhance the inner temperature of the filament region, and thus avoiding large local electric field[34]. The conductive metal oxide layer also helps to reserve plenty of oxygen ions, which improves the analogue behaviour during RESET process. The robust analogue switching behaviour with good cell-to-cell uniformity (Supplementary Note 1) benefits a lot from the utilizing of HfAl$_y$O$_x$ switching layer, since HfO$_2$ is well-known as phase-stable. And the HfO$_x$/AlO$_y$ laminate structure is leveraged to control the generation of oxygen vacancies in such a RRAM cell design. The ratio between HfO$_2$ and Al$_2$O$_3$ is well adjusted during fabrication process and optimized as 3:1. This structure design shows a better analogue performance compared with TiN/TaO$_x$/HfO$_2$/TiN (Supplementary Note 2).

The 1T1R structure is used to further improve the bidirectional analogue switching performance and uniformity. Compared to the two-terminal RRAM cell, the three-terminal 1T1R cell improves the controllability of continuous weights tuning at array level since the compliance current controlled by the transistor's gate voltage can significantly suppress the overshoot and feedback effects during SET process. In addition, exploiting transistors could guarantee persistent scaling-up of array scale by eliminating the sneak path and avoid several bottlenecks of analogue RRAM array.

Figure 3a shows the smooth and symmetrical I–V curves of the optimized 1T1R cell. A 40 times window is exhibited using a quasi-DC sweep. The elimination of the abrupt conductance transition at both SET and RESET processes enables bidirectional continuous conductance change for weight update. Typical analogue behaviour under identical pulse train during SET and RESET processes is shown in Fig. 3b,c, respectively, showing that the conductance can be modulated by applying identical voltage pulses (conductance changes under continuous SET and RESET pulse cycles is shown in Supplementary Fig. 8). This remarkably simplifies the update strategy and control circuits. Similar analogue behaviours can be observed under different pulse conditions for a wide range of pulse amplitude and duration (see Supplementary Note 1).

To further suppress the influence of the slight resistance variation across cells in the array, a write-verify programming scheme that is in accordance with equation (2) is proposed and experimentally compared with the scheme without write-verify that implements equation (3). The write-verify flow is shown in Supplementary Fig. 9. During the weight update phase of each learning iteration, identical voltage pulses are applied to the 1T1R cell to increase (or decrease) the cell conductance, until the conductance is larger (or smaller) or equal to the target value[35], based on equation(2). Hence the final synapse weight only slightly deviates from the target in most of cases. In contrast, without write-verify, only one SET (or RESET) pulse is applied on the selected 1T1R cell to increase (or decrease) the conductance without checking whether it reaches the target value or not. Avoiding the write-verify step simplifies the control circuit since it is not necessary to calculate the specific analogue value of the error between the target weight and the current weight, but it may slow down the convergence due to cycle-to-cycle and device-to-device variations. Figure 3d,e specify the waveforms during the SET process for the scheme with and without write-verify. Similar RESET waveforms are illustrated in Supplementary Fig. 10 and are applied in parallel row by row as well.

The switching window and conductance modulation steps depend on the pulse width and the pulse amplitude, which leads to a trade-off between the learning accuracy and the convergence speed. The effects of the pulse condition on the training process are shown in Fig. 3f,g. We can see the opposite trend that a higher pulse amplitude requires less number of pulses but results in a larger deviation from the target, creating a trade-off between accuracy and speed. When the pulse amplitude is < 1.5 V, the device conductance is not able to reach the higher conductance range (for example, > 10 μS). This implies another trade-off between the conductance modulation range and the accuracy, which is detailed in Methods section 'Device performance during write-verify RESET process'. The operation conditions should be carefully optimized according to application at hand as well as speed, energy and accuracy requirements: for example, lower amplitude and shorter duration could be employed to increase modulation accuracy for both training rules at the expense of speed. Similar measurement is conducted during write-verify SET process and the result is shown in Methods section 'Device performance during write-verify SET process'. The SET and RESET operation conditions with $V_{wl} = 2.3$ V, $V_{bl} = 2.1$ V (50 ns) and $V_{wl} = 8.0$ V, $V_{bl} = 2.0$ V (50 ns) provides a reasonable balance between accuracy and speed and hence are used in the following experiments.

**Grey-scale face image classification**. The optimized 1024-cell-1T1R array is used to demonstrate face classification by the neuromorphic network. All the 1T1R cells are programmed to an initially state around 40 μS. Even with slight device variations, the system works smoothly under both operation schemes. The network converges after 10 iterations for the write-verify operation scheme, while for the scheme without write-verify, the network converges after 58 iterations. Figure 4a,b reveal the progress of the training process when identifying the face of the first person. The trace of conductance evolution in single RRAM cell is shown in Supplementary Note 3. The final conductance distribution and visual map diagram are presented in Fig. 4c,d. Conductances are normalized as integers from 0 to 255 in the map. The detailed process for the faces of other two people are provided in Supplementary Figs 15 and 16. Furthermore, another two demonstrations are conducted, one starting from a tight low conductance distribution around 4 μS and another proceeding from a wide conductance distribution state. Both succeed to converge (see Supplementary Note 4). The initial distribution state hardly affects the convergence of the training.

Two sets of patterns are used in the test process. One set contains 24 images (Supplementary Fig. 19) in the Yale Face Database for these three persons (not shown during training). The other set consists of 9,000 patterns constructed by introducing noise to the training images. Noise patterns are

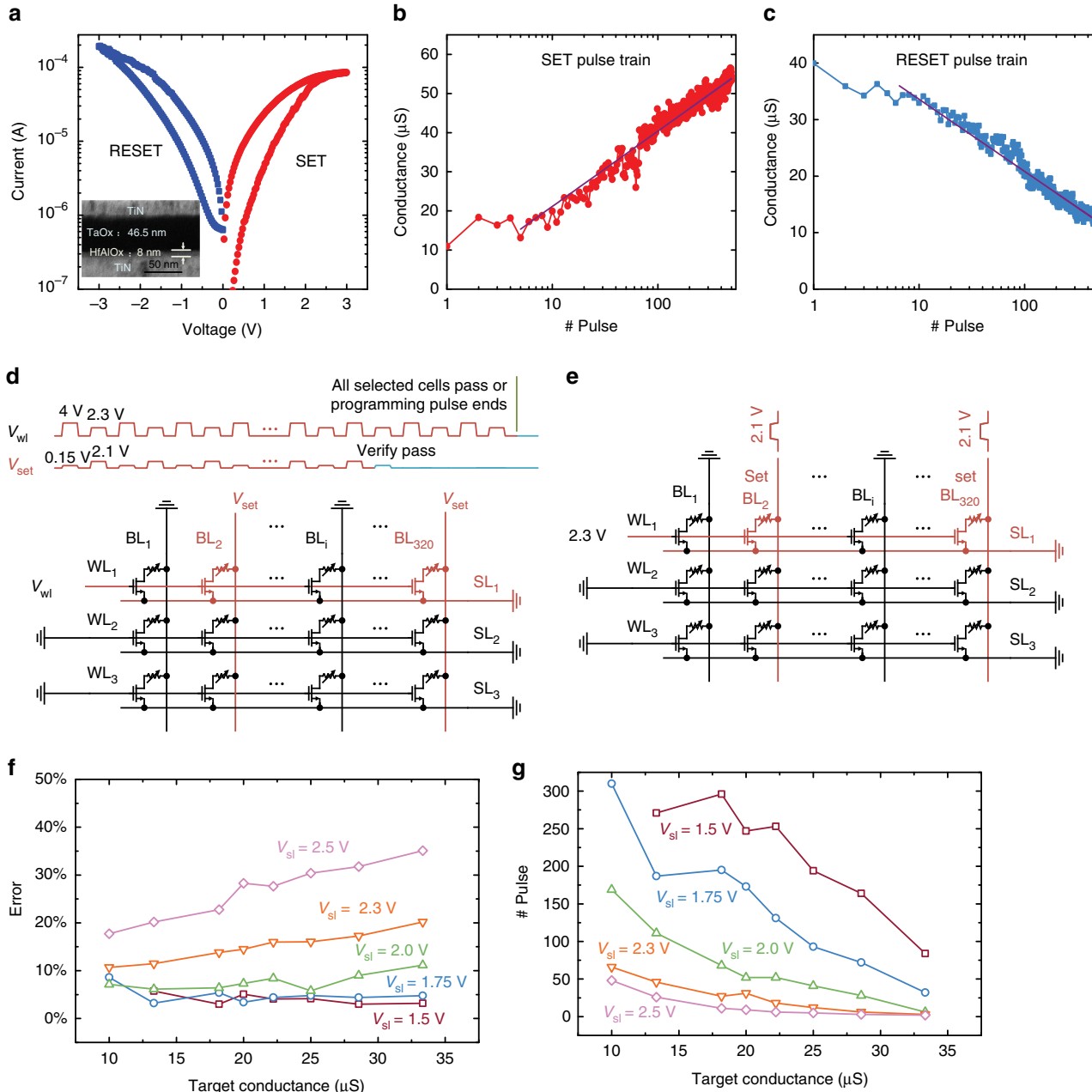

**Figure 3 | The performance of the optimized device and two examples for programming.** (**a**) Typical *I–V* curve of a single 1T1R cell for a quasi-DC sweep, the gate voltage is 1.8 and 8 V during SET and RESET process, respectively. Inset is a transmission electron microscope (TEM) image of the RRAM device. (**b**) An example of the typical continuous conductance tuning performance under an identical pulse train condition during SET process, along with the fitting curve. $V_{wl} = 2.4$ V, $V_{bl} = 2.0$ V (50 ns), $V_{sl} = 0$ V. (**c**) Tuning performance during RESET operation, along with the fitting curve. $V_{wl} = 8$ V, $V_{bl} = 0$ V, $V_{sl} = 2.3$ V (50 ns). (**d**) An example of the SET programming waveform applied on the first row to adjust the weight, following write-verify scheme. (**e**) Waveforms for programming without write-verify. (**f**) The precision measurement result during RESET process using verified pulse train with different amplitudes. *y* Axis represents the final conductance accuracy (the difference between the target conductance and the measured conductance over the target conductance) after programming from a same initial state 40 µS. (**g**) *y* Axis represents the number of pulses needed to reach the target conductance from the same initial state 40 µS. These curves show the relationship of tuning speed with respect to different programming pulse amplitudes.

generated by randomly choosing some pixels and assigning them a random value. One thousand different patterns are generated from each training image, in which different numbers of noise pixels (1–100) are introduced. Three noise patterns are presented in Fig. 5a. For the test patterns without noise, 2 out of the 24 patterns are misclassified using the write-verify scheme; whereas 3 patterns are misclassified using the scheme without write-verify, as Fig. 5c shows. This is close to the 22/24 accuracy with the standard computing system. The real-time changes of

the misclassification rate under the two schemes during training are given as well (Supplementary Note 5). Figure 5b shows the recognition rate on the 9,000 augmented noisy test patterns. It is shown that scheme with write-verify presents a much lower misclassification rate for the entire set of testing patterns. This trend indicates that more noise pixels lead to a lower recognition rate. The average recognition rate on the total 9,000 augmented noisy test patterns is 88.08% and 85.04% for the write-verify and without write-verify methods, respectively, slightly decreasing

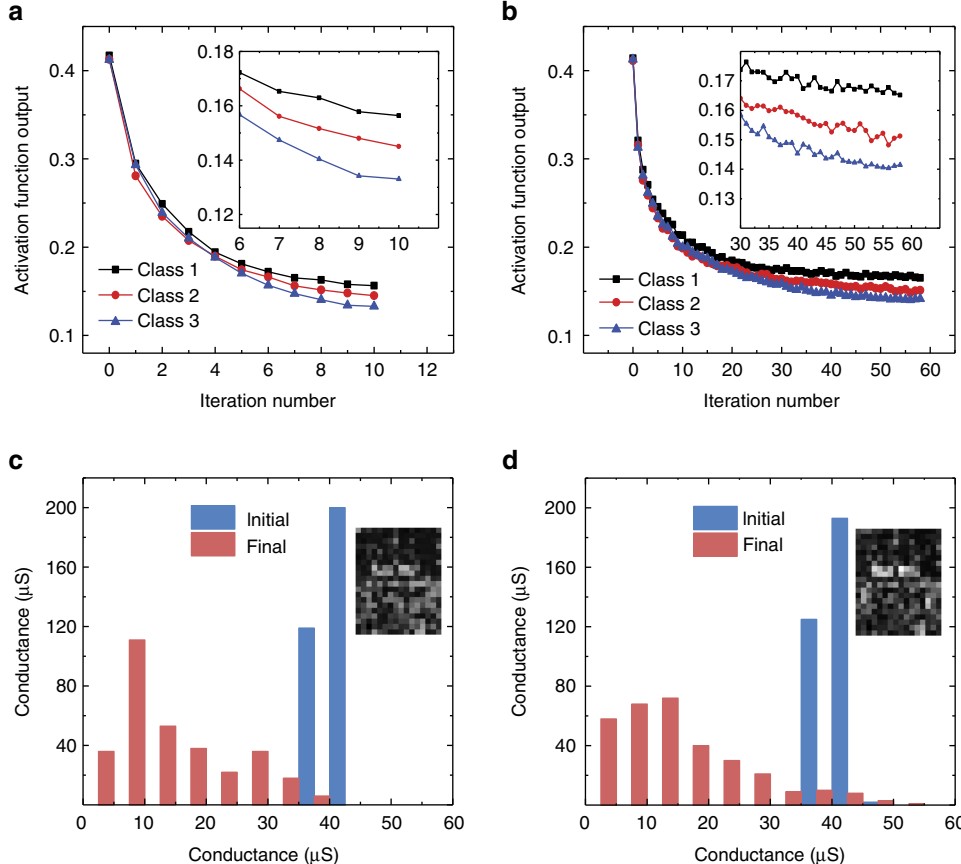

**Figure 4 | The training process of the experimental demonstration.** (**a**) The activation function output value of the first class versus the iteration number using the write-verify scheme. The inset figure zooms in the several last steps. (**b**) The training process for programming without write-verify. (**c**) The initial and final conductance distribution comparison of the first row when updating with write-verify. Inset shows the final conductance map. (**d**) The conductance distribution of the first row and the conductance map for the case without write-verify.

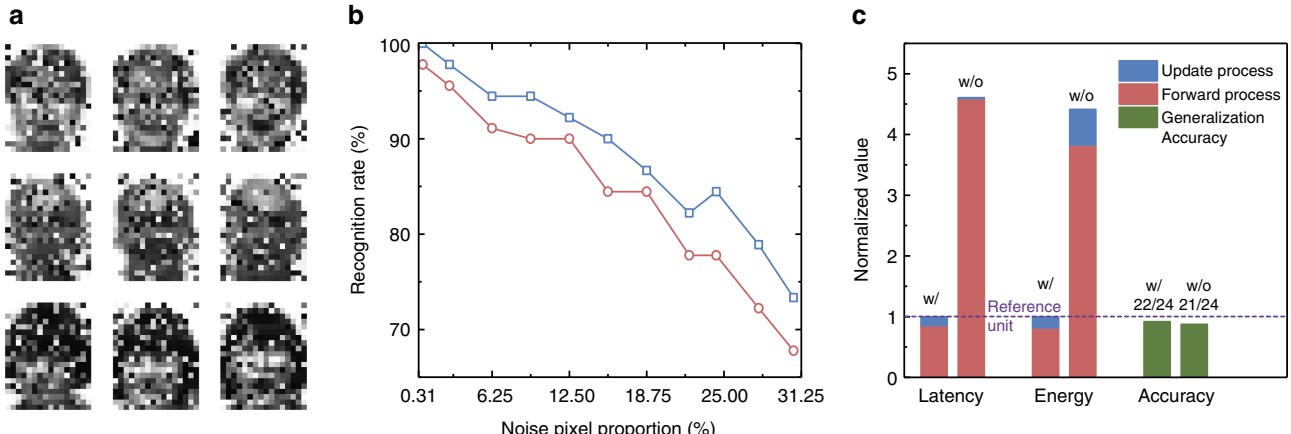

**Figure 5 | The test result with latency and energy comparison.** (**a**) A standard example of the constructed test patterns with 100 noise pixels with respect to each training image in Fig. 2c. (**b**) The recognition rate curve of two programming strategies during the test. The x axis represents the amount of noise. (**c**) The comparison between with (W/) and without (W/O) schemes in terms of latency and energy consumption during training process and testing recognition rate in the normalized format.

compared with the 91.48% recognition rate by standard method. Total latency and energy consumption comparisons during the training process are presented in Fig. 5c. These data are acquired from experimental measurements. As the input is encoded by the pulse number, which has the maximum value of 255, the contribution to latency and energy consumption of inference phase is quite large. During weight update phase, the total latency

and energy is actually 422.4 μs and 61.16 nJ for the write-verify scheme from the beginning to the end, whereas the corresponding speed and energy using the scheme without write-verify is 34.8 μs and 197.98 nJ, respectively. Considering that the scheme with write-verify needs more programming pulses at each epoch, the write-verify scheme requires relatively longer latency during weight update phase. However, it performs better when taking

energy consumption into consideration, and this is mainly due to the lower number of iterations required. Besides, the total latency and energy during the entire training process can benefit a lot from the decrease of the number of iterations because the major latency and energy consumed by inference task could be suppressed. Therefore, although the scheme without write-verify simplifies the update operation, the scheme with write-verify has superior performance of recognition accuracy, total latency and energy consumption using the same pulse amplitude and width.

**Analogue RRAM array enables lower energy consumption.** The energy consumption of the same network executed on conventional computing platforms are estimated and compared with the hardware used in this experiment. The average energy consumption leveraging Intel Xeon Phi processor[36] with a hypothetical off-chip storage is $1,000 \times$ larger than this work, given that the average energy consumption is around 30 nJ (33 and 25 nJ for scheme with write-verify and without write-verify, respectively) at every epoch for the same classification task. Energy consumed due to off-chip non-volatile storage access dominates in the case of off-chip non-volatile memory, as the write energy of a 2 KB page size is 38.04 $\mu$J per page using NAND flash[37]. For this network, the weight matrix is roughly 2 KB for 16-bit weights. If a hypothetical hardware with Intel Xeon Phi processor where digital RRAM is integrated on-chip is assumed, the energy consumption is roughly 703 nJ per epoch, which is $20 \times$ larger than reported in this paper using on-chip RRAM analogue weight storage. The bidirectional analogue RRAM array realizes remarkable energy consumption saving and reaches a comparative accuracy during this experimental demonstration. It is important to note that these energy values only include the energy consumption for synaptic operations (reading synapses and updating synapses) and not the computation within the neurons (see Supplementary Note 6 for details).

## Discussion

In summary, a neuromorphic network is developed using a bidirectional analogue 1024-cell-1T1R RRAM array. The optimized RRAM metal oxide stack (TiN/TaO$_x$/HfAl$_y$O$_x$/TiN) exhibits gradual and continuous weight change. Based on this device technology, an integrated neuromorphic network hardware system is built and trained online for grey-scale face classification. Both with and without write-verify operation schemes are studied for the neuromorphic network and they achieve a relatively high recognition rate after converging, that is, 22/24 and 21/24, respectively. There is trade-off between these two schemes. The scheme with write-verify shows a much better approach providing $4.61 \times$ faster converging speed, $1.05 \times$ higher recognition accuracy and $4.41 \times$ lower energy consumption, whereas the scheme without write-verify simplifies the operation to a great degree. This integrated neuromorphic network hardware system has remarkable energy consumption benefit compared to other hardware platforms. The resistive switching memory cell can be scaled down to 10 nm (ref. 33), which provides around $10^{11}$ synapses per cm$^2$. With further monolithic integration with neuron circuits, more complex deep neural networks and human-brain-like cognitive computing could be realized on a small chip. Meanwhile, it has to be noticed that the related accuracy, speed and power are all important for the actual application[38]. To achieve a comparable classification accuracy on larger network as the state of art and realize the superiority on power and speed simultaneously, there are many technical issues to be solved. Both experimental and simulation efforts should be paid on the device optimization,

algorithm modification, operation strategy improvement and system architecture design[39].

## Methods

**RRAM stack and fabrication process.** The metal-oxide-semiconductor field-effect-transistor circuits are fabricated in a standard CMOS foundry. The technology node is 1.2 $\mu$m. The CMOS circuitry works as the WL decoder and cell selector. The RRAM devices are formed on the drain of the transistors by using the following processes (Supplementary Fig. 2). The HfO$_2$/Al$_2$O$_3$ multilayer structure is deposited on the TiN bottom electrode with atomic layer deposition method by repeating HfO$_2$ and Al$_2$O$_3$ cycles at 200 °C periodically. For each period, three cycles of HfO$_2$ and one cycles of Al$_2$O$_3$ are deposited. The thickness of one atomic layer deposition cycle of both HfO$_2$ and Al$_2$O$_3$ is around 1 Å. The final thickness of the HfAl$_y$O$_x$ layer is about 8 nm. Then a 60 nm TaO$_x$ capping layer that acts as an in-built current compliance layer and oxygen reservoir is deposited by physical vapour deposition method. The top electrode TiN/Al are deposited by reactive sputtering and electron beam evaporation, respectively. Finally, the top Al pad is patterned by dry etching with Cl$_2$/BCl$_3$ plasma.

**Write-verify programming method.** Two programming schemes, one with write-verify while the other without, are proved at array level. The scheme with write-verify is described in Supplementary Fig. 9. Target conductance values are send to the Tester (Supplementary Fig. 3) in each learning iteration and multiple electrical pulses are applied to the 1T1R cell to increase (decrease) the conductance, until the conductance is larger (smaller) or equal to the target values. Finally, the cell conductance slightly deviates from the target in most of the cases.

**Device performance during write-verify RESET process.** Pulse amplitude and pulse width highly effect the cell performance according to Supplementary Figs 4 and 6. Meanwhile, we can conclude from Fig. 3f,g of the main text that there is a tradeoff between tuning speed and tuning accuracy. Further, Supplementary Fig. 11 implies that the conductance modulation range must be considered when determining the pulse condition.

During the experiment of Fig. 3f,g and Supplementary Fig. 11, a sequence of identical RESET pulses with write-verify are applied to examine how varied pulse amplitudes affect weight adjustment. The raw data are statistically averaged over 32 random chosen cells under 3 repeated procedures to get rid of device variances. The procedure starts with precisely initializing cell conductance at 40 $\mu$S (25 k$\Omega$). Then a specified pulse train is applied to tune cell conductance to a certain target value. The refined conductance value and total pulse number when write-verify passes are recorded. The programming pulse width is 50 ns, and the gate voltage $V_{wl}$ is 8 V. The BL is grounded and the pulse number limitation is set to 500. Several trials are conducted during each test, tuning the 32 cells' conductance to 33.3 $\mu$S (30 k$\Omega$), 28.6 $\mu$S (35 k$\Omega$), 25 $\mu$S (40 k$\Omega$), 22.2 $\mu$S (45 k$\Omega$), 20 $\mu$S (50 k$\Omega$), 18.2 $\mu$S (55 k$\Omega$), 13.3 $\mu$S (75 k$\Omega$) and 10 $\mu$S (100 k$\Omega$).

**Device performance during write-verify SET process.** Similar measurement is conducted during write-verify SET process to see how pulse amplitude affect conductance modulation precision, modulation pass rate and modulation speed.

During the test, a sequence of identical SET pulses with write-verify are applied to examine how pulse amplitudes affect weight adjustment. The raw data are statistically averaged over 32 random chosen cells under 3 repeated procedures to get rid of device variances. The procedure starts with precisely initializing cell conductance at 4 $\mu$S (250 k$\Omega$). Then a specified pulse train is applied on BL to tune cell conductance to a certain target value. The refined conductance value and total pulse number when write-verify pass are recorded. The programming pulse width is 50 ns, and the gate voltage $V_{wl}$ is 2.8 V. The SL is grounded and the pulse number limitation is set to 300. Several trials are conducted during each test, tuning the 32 cells' conductance to a same target conductance target set as in the RESET test, that is, 33.3 $\mu$S (30 k$\Omega$), 28.6 $\mu$S (35 k$\Omega$), 25 $\mu$S (40 k$\Omega$), 22.2 $\mu$S (45 k$\Omega$), 20 $\mu$S (50 k$\Omega$), 18.2 $\mu$S (55 k$\Omega$), 13.3 $\mu$S (75 k$\Omega$) and 10 $\mu$S (100 k$\Omega$). The results are shown in Supplementary Fig. 12. We can conclude that there is a tradeoff between tuning speed, conductance modulation range and tuning accuracy when determining the SET pulse conditions.

**The perceptron network.** A one-layer perceptron is adopted for this hardware system demonstration, and the schematic diagram is shown in Supplementary Fig. 1. The perceptron model is used to classify each pattern to three categories. This schematic illustrates how to map the network to the proposed 1T1R structure, that is, the input of preneuron layer, adaptable synaptic weight and the weighted sum output of postneuron layer are in accordance with the pulse input from BL, 1T1R cell conductance and current output through SL separately. The nonlinear function 'tanh' is regarded as the activation function here.

**Unseen test images from the Yale Face Database.** We have obtained full permissions to use the images from Yale Face Database and are compliant with Yale's policy of reuse/use of these images (http://vision.ucsd.edu/content/yale-face-

database). The total 9 training images are presented in Fig. 2c, the other 24 cropped and down-sampled face images from the Yale Face Database are used to evaluate the perceptron's generation ability, as shown in Supplementary Fig. 19.

**Test platform and the hyper-parameter values.** As is mentioned in the main text, the weights are implemented using the 1,024-cell-1T1R array, and the nonlinear activation function tanh with respected to the SL current is implemented by the software. The control instructions are sent to the external equipment to generate practical programming pulses side by side. All these in union work automatically. The diagram of this platform is shown in Supplementary Fig. 3.

A fitted behaviour RRAM model is extracted from experiment data and used for the simulation to decide the hyper-parameters in Fig. 2a. Eventually $\beta$ is defined as $1.5\,A^{-1}$, and $\eta$ equals to 1. Apart from these, the target value of the activation function $f^t$ is 0.3 for the right class and 0 for other wrong classes during training process.

**Data availability.** The data that support the findings of this study are available from the authors upon reasonable request; see Author contributions section for specific data sets.

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

## Acknowledgements

We thank Professor Shimeng Yu of Arizona State University for the valuable discussions. We acknowledge the use of the Yale Face Database. This work is supported in part by the Beijing Advanced Innovation Center for Future Chip (ICFC), National Key Research and Development Program of China (2016YFA0201803), National Hi-tech (R&D) Project of China (2014AA032901), and NSFC (61674089). S.B.E. and H.-S.P.W. are supported in part by the National Science Foundation Expeditions in Computing (Award no. 1317470) and member companies of the Stanford SystemX Alliance.

## Author contributions

P.Y., H.W., B.G. and S.B.E. designed the research and conceptualized the technical framework. P.Y., X.H. and W.Z. performed the experiments. P.Y., Q.Z. and S.B.E. contributed to the simulation. P.Y., B.G. and H.-S.P.W. contributed to the paper writing. All authors discussed and reviewed the manuscript. H.W. and H.Q. were in charge and advised on all parts of the project.

## Additional information

**Competing interests:** The authors declare no competing financial interests.

