## [Peer Review File · Nature Communications]

Reviewers' comments:

Reviewer #1 (Remarks to the Author):

The authors present a very exciting face recognition problem based on RRAM devices. This paper is certainly of great interest for a broad audience and represent a significant milestone for neuromorphic engineering. The paper is well organized and the experiment are well exposed and performances analyzed. The paper meets the criterion for publication in Nature Communication.

Some comments that should be address before publication:

- The learning is presented in all the case with devices starting from a tight distribution around the ON state. Consequently, it seems that devices only experience a reset transition during learning.

1/ Could the authors add the evolution of few randomly selected weights during learning?

2/ What happens for learning if the RRAM distribution start from the OFF state? Is the learning still successful and how energy consumption evolved?

3/ What happens if the learning start from a large distribution of weight (equivalent to the one reach at the end of learning in the paper)? This point should be check in order to address the robustness of learning. Is the system able to perform many successive learning or does it need to be "refresh" before learning a new task?

4/ In the SI, could the authors precise th eCMOS technology used (node) and give more details about what is done by the CMOS circuitry.

5/ regarding the energy consumption comparison with standard systems, is the estimate really fare? First, 16 bit is certainly more than what is done by the RRAM. Secondly, Multiply operation in conventional system also include signal generation and overall system operation. In the case of RRAM, the energy evaluation seems to include only device switching and not all the associated circuitry. Could the authors address this point in more detail?

Reviewer #2 (Remarks to the Author):

The authors fabricated a memristive neural network based on 1024-cell RRAM array. A face recognition was performed showing about 90% recognition rate. Overall, the paper is interesting and results are sound. The neural network application is quite standard.

It should be noted that an integrated memristive neural network of 9 memristors was demonstrated by Strukov's group in 2015 [Nature 521, 61-64 (2015)]. Consequently, the present work represents rather an important technological step than a scientific one. In my opinion, the manuscript is suitable for publication in Nature Communications.

The abstract and introduction are a little bit overhyped. It's true that the memristive NN requires less power, but let's also compare the face recognition accuracy. Therefore, as a mandatory revision, the recognition rates for the standard and demonstrated approaches should be given in the abstract.

I like that the authors provide the individual memristor response in the Supplementary materials (Fig. 3). Hopefully, the human synapses are less noisy.

Reviewer #3 (Remarks to the Author):

This manuscript describes an experimental implementation of the on-chip training of a very small, single-layer perceptron neural network on a toy-size problem. The in-situ training is described, generalization and noisy inference is tested, and network accuracy is shown. The authors do share a large amount of data illustrating the high variability of their devices, which is good. Authors also focus overly much on the power advantages of such an online array, which of course are large -- yet do not provide useful information to other experts in the field, vis-a-vis: If we or they built a large array of these devices and used it to train a multi-layer neural network on a non-toy problem, would there be any chance of it actually working (e.g., converging to acceptably-interesting accuracies within any reasonable time)?

I recommend the authors go back and attempt to answer the question above using simulation of neural networks as informed by their extensive data on device-to-device variability and conductance response and provide the answer in their revised paper. Also, since I have very good reason to believe the answer to my question above is "No, it won't work with these devices," then authors will furthermore need to describe for the reader how much these devices need to be improved (in order to make non-toy DNN work), whether this is feasible, and how this could potentially be done.

I recommend reject and mandatory revision before resubmission.

Specific comments:

1) The authors include the usual hand-waving rhetoric about the glories of neuromorphic computing, but of course low energy computation is completely useless if you are not obtaining a result of relevance. In this context, future neuromorphic systems built around such devices **MUST** be able to train the exact same size (or bigger) networks as are currently trained by GPUs, **MUST** obtain similar classification accuracies by the end of training, and most likely **MUST** provide a speedup advantage **IN ADDITION TO** lower power.

Yes, there is a niche market for solutions that train DNN at lower power to sufficient accuracy but more slowly than GPUs, but it is a fairly small niche. There is **NO** niche for solutions that fail to achieve similar classification accuracies on non-toy-size DNN. There is zero mention or acknowledgement of this reality in the current intro to this paper, and this must be fixed.

2) Authors claim "this is the first experimental demonstration of such a significant cognitive task." I get tired of papers that are explicitly built so that one can claim they are "the first," leaving all the actual hard work of taking something from first demonstration to usefulness to someone else. In this case, I cannot agree that the network shown here can be considered "significant." This network and dataset is far too small to allow any predictive (or even indicative) power for larger non-toy networks. One can say that MNIST itself is also far smaller in size and easier in difficulty than CIFAR or ImageNet, but if a memristor-based solution were able to deliver identical-accuracy-

performance

on MNIST, I would have to claim that that accomplishment would be a major step towards the demonstration of

NVM-based neuromorphic systems. The present network is far far smaller and much easier -- to the extent

that success on this toy problem is indicative of nothing.

3) There is a major bait-and-switch between the aspect ratio of the experimental array (described as having 128 rows and 8 columns) and the logical array needed for the network of interest (described

as having 3 rows and 320 columns). Thus statements like "A fully parallel read operation" is clearly a bald-faced lie, since there is absolutely no way the authors can experimentally be inputting all 320 columns at the same time given the dimensions of the physical array. Statements like this will need to

be fixed to reflect the actual realities of the experiment as it was actually performed.

4) Ways in which this experiment is a toy experiment

- It is not that the images are too small - 20x20 is not far off in size from MNIST. The problem is that there

are so few classes, which appear to be cherry-picked from the dataset in order to maximize the distance between

the classes and make the problem easier.

- Batch-based processing (e.g., going through the whole dataset and THEN updating all the weights) has long been

abandoned by the AI community. IF you wanted to do this for a large dataset, you would take a large hit in

convergence rate, plus you must describe how this data will be stored locally (or include the power for aggregating

this data offline in your power and latency assessment).

- The convergence of this learning appears to be so rapid that the authors are able to almost completely avoid having

to use any SET pulses during the experiment. This will most definitely not be the case for any non-toy experiment.

(Authors should show the histogram of desired conductance changes during training, so readers will know what the

balance between requests-for-lower-conductances and requests-for-higher-conductances).

- Because the dataset is toy-size, authors probably terminated training too early, thus losing the opportunity for

further improvements in generalization performance or classification of noisy inputs.

- The hundreds of write-read-verify cycles required for these devices are not going to scale well for a non-toy

problem, without some trick which allows you to do this extremely infrequently and/or efficiently.

5) Other comments:

- If authors are going to focus on power to this extent, they must include the power required to implement their tanh()

function, to compute the delta values after each example, and to aggregate and store the weight updates during their

batch-based programming.

6) The RRAM devices do seem to show a nice gentle behavior. This reviewer is impressed at the slick

usage of a logarithmic horizontal axis to make the conductance response "look" linear, when in actuality it is quite far from linear. That said, the supplementary data in S3 S4 and S5 (showing just how bad the variability between these devices really is) is greatly appreciated by the reviewer. After seeing this data, I was amazed that even this toy experiment worked at all, until I realized that the toy database only required RESET pulses, and that in the verification of your experiment, you were willing to wait as long as it took, even if that meant many hundreds of programming pulses.

One comment on Figure 3B,C -- it is disconcerting that the vertical axes of B and C are not the same scale. You should either start your RESET characteristic at the same peak value that the SET characteristic terminates at, OR at least make the two plots have the same vertical scale so that the reader can immediately observe that the SET and RESET characteristics are NOT completing each other.

Assuming that there were even any SET pulses needed during the "training" of this toy problem, it would be good to show the "SET" versions of Figures 3F,G.

7) The results with classification of noisy training data and with the test data (non-training data) are appreciated, and help improve the quality of the paper. Two points here:

- since the 9000 noisy images are noise added to the training images, they should NOT be referred to EVER as "test patterns" (see line 269). It is VERY important that it be 100% clear this is noise on already-seen training images, NOT on never-before-seen test data.

- to differentiate the statistically relevant percentage numbers in Figure 5b (out of 9000 noisy images) from the generalization "percentages" in Figure 5c, please label the green bars as generalization accuracy and include the exact fraction of correct generalized images (e.g., 22/24, etc.) as well as the percentage (to make it clear these "percentages" are accumulated over a very small number of instances).

Authors take great pains to laud the great power of experimental results, and similarly spend significant effort disparaging simulation-based efforts (including Ref 36 and the relevant portions of Ref 19, which also included a MUCH larger "first ever" experiment than the present manuscript). However, this reviewer would claim that simulations -- especially ones that are done well enough that they can project accurately for the field whether the devices at hand will succeed or will fail at tasks of actual

interest -- are much more useful for the field than experimental demonstrations at toy scale without such simulations. In most fields, toy experiments have little forward predictive power for tasks of actual interest -- for neural networks, they have zero forward predictive power.

That said, it would not take much for the authors of this manuscript to take their extensive experimental data and to project through simulation whether these devices as they stand (nonlinear conductance response AND massive device-to-device variability) would suffice for even easy non-toy problems like MNIST. Such simulations would also determine whether the nonlinear conductance response itself were invalidating --- e.g., if every single device in a large array functioned exactly like the device described by Figure 3B,C with ZERO variability, even in that ideal case would one have a shot at training MNIST successfully?

This is the most important and relevant question, which remains unanswered (or even acknowledged) by the present manuscript. This is why the manuscript should not be accepted as it currently stands.

Without this kind of information, the reviewer cannot agree that "these experimental results pave the way" towards the desired energy-efficient neuromorphic systems. Instead, this manuscript is merely yet another "first to show yet more useless results" paper.

REVIEWERS' COMMENTS:

Reviewer #1 (Remarks to the Author):

The authors answered all the points raised by the reviewer and consider all the suggestions comments raised by the other reviewer. The paper is acceptable for publication in nature communication

Reviewer #2 (Remarks to the Author):

The revised version of the manuscript takes into account all my suggestions and comments. Therefore, I believe that it is suitable for publication in Nature Communications.

Reviewer #3 (Remarks to the Author):

Authors responded to many of my points, and new data described that addressed a number of my complaints.

I can recommend this revise manuscript for acceptance.

Reviewer 1

We would like to thank the reviewer for the comments. According to the comments, we have made the revisions on the manuscript. Here is the point-by-point reply to the reviewer's Comments.

Comment 1:

The authors present a very exciting face recognition problem based on RRAM devices. This paper is certainly of great interest for a broad audience and represent a significant milestone for neuromorphic engineering. The paper is well organized and the experiment are well exposed and performances analyzed. The paper meets the criterion for publication in Nature Communication. Some comments that should be address before publication:

The learning is presented in all the case with devices starting from a tight distribution around the ON state. Consequently, it seems that devices only experience a reset transition during learning. Could the authors add the evolution of few randomly selected weights during learning?

Response:

We appreciate your positive and insightful comments. The question is very interesting. We checked the experiment data and found that 19.3% of the devices experienced SET transition during learning under the write-verify scheme. The weights evolution trace of ten randomly selected cells are shown below (left two columns, in red), together with another ten randomly selected cells from the rest cells (right two columns, in blue) which merely experienced RESET transitions.

Fig. R1. Conductance evolution of 20 randomly selected RRAM devices during learning process under the write-verify scheme. The figures with red lines indicates the cells which experienced SET processes. And the figures with blue lines indicate the cells which merely experienced RESET processes.

Under without write-verify scheme, the percentage of the devices which experienced SET during learning is 14.6%. The weights evolution trace of 20 randomly selected cells is presented below.

Fig. R2. Conductance evolution of 20 randomly selected RRAM devices during learning process under the without write-verify scheme. The figures with red lines indicates the cells which experienced SET processes. And the figures with blue lines indicates the cells which merely experienced RESET processes.

Location of Change:

“Grey-scale face image classification” Section in the main text, from line 256 to line 257.
 Section 11 in Supplementary Information, from line 158 to line 164.

Comment 2:

What happens for learning if the RRAM distribution start from the OFF state? Is the learning still successful and how energy consumption evolved?

Response:

We carried out experiments where cells start from OFF states. We found the learning is still successful. The energy consumption per iteration is 23 nJ under the write-verify scheme and 21 nJ under the without write-verify scheme which are very close. The energy consumptions for the learning starting from OFF state are slightly smaller than the energy consumption if learning starts from the ON state.

The comparisons of initial and final conductance distribution are shown below.

Fig. R3. The comparisons of initial and final conductance distribution under the proposed two updating schemes starting from the OFF state. The three figures above show the comparative distribution of 1st class, 2nd class and 3rd class under write-verify scheme respectively. The three figures below show the comparative distribution of 1st class, 2nd class and 3rd class under without write-verify scheme respectively.

Location of Change:

“Grey-scale face image classification” Section in the main text, from line 260 to line 264.
Section 13 in Supplementary Information, from line 172 to line 179.

Comment 3:

What happens if the learning start from a large distribution of weight (equivalent to the one reach at the end of learning in the paper)? This point should be check in order to address the robustness of learning. Is the system able to perform many successive learning or does it need to be "refresh" before learning a new task?

Response:

We carried out new experiments to check the impact of initial weight distribution starting from a wide distribution that half weights were randomly SET to ON state while the others

were randomly RESET to OFF state. Since the device has bi-directional analog switching behavior, it does not matter what the initial conductance distribution is. The learning process converged as well. The energy consumption is 24 nJ under the write-verify scheme and 23 nJ under the without write-verify scheme. The initial and final conductance distribution comparison of the total weights are shown below.

Therefore, we can conclude that this learning system possesses sufficient robustness and does not need “refresh” before a new task.

Fig. R4. The comparisons of initial and final conductance distribution under the proposed two updating schemes starting from a wide-distribution state. The three figures above show the comparative distribution of 1st class, 2nd class and 3rd class under write-verify scheme respectively. The three figures below show the comparative distribution of 1st class, 2nd class and 3rd class under without write-verify scheme respectively.

Location of Change:

“Grey-scale face image classification” Section in the main text, from line 260 to line 264.
 Section 13 in Supplementary Information, from line 172 to line 179.

Comment 4:

In the SI, could the authors precise the CMOS technology used (node) and give more details about what is done by the CMOS circuitry.

Response:

We added some related statement in the Supplementary Information.

Location of Change:

Section 2 in Supplementary Information, from line 41 to line 42.

Comment 5:

Regarding the energy consumption comparison with standard systems, is the estimate really fair? First, 16 bit is certainly more than what is done by the RRAM. Secondly, multiply operation in conventional system also include signal generation and overall system operation. In the case of RRAM, the energy evaluation seems to include only device switching and not all the associated circuitry. Could the authors address this point in more detail?

Response:

First, we take advantage of the analog switching behavior of the RRAM weights which is different from digital bit quantization. In the manuscript, we compared the energy consumption with a standard CPU system assuming 16-bit weight. If 8-bit weight was used, we still got remarkable energy saving. The energy consumption within the analog synapses for each iteration is 1,000x lower compared to an implementation using an Intel Xeon Phi processor with off-chip memory, and is 10x lower compared to an implementation using an Intel Xeon Phi processor with a hypothetical integrated on-chip digital RRAM.

Secondly, in this paper, we reported on the energy consumption on operation of the 1T1R array which includes the multiply operation and weight updating process. In comparison, we also only considered the energy consumption of the similar operations for a standard system. The energy for the operations beyond the multiply operation and weight updating process was not taken into consideration for comparison. Specifically, in this case, we did not take the signal generation into consideration when estimating the energy consumption of the multiply operation for a standard system. The data used for estimation was obtained from Ref. 36 for register-to-register multiply operation.

The detail information about the energy consumption calculation could be found in Supplementary S16.

Location of Change:

Section 16 in Supplementary Information, from line 191 to line 198 and line 204.

Reviewer 2

We would like to thank the reviewer for the comments. According to the comments, we have made the revisions on the manuscript. Here is the point-by-point reply to the reviewer's Comments.

Comment 1:

The authors fabricated a memristive neural network based on 1024-cell RRAM array. A face recognition was performed showing about 90% recognition rate. Overall, the paper is interesting and results are sound. The neural network application is quite standard.

It should be noted that an integrated memristive neural network of 9 memristors was demonstrated by Strukov's group in 2015 [Nature 521, 61-64 (2015)]. Consequently, the present work represents rather an important technological step than a scientific one. In my opinion, the manuscript is suitable for publication in Nature Communications.

The abstract and introduction are a little bit overhyped. It's true that the memristive NN requires less power, but let's also compare the face recognition accuracy. Therefore, as a mandatory revision, the recognition rates for the standard and demonstrated approaches should be given in the abstract.

Response:

We appreciate your positive and insightful comments. We agree that the accuracy is an important metric. We added the recognition rate comparison between standard method and the experimental demonstration in the abstract, introduction, and "Grey-scale face image classification" section. The convergence is triggered as long as the perceptron recognizes all of the training images. The following table shows the detail recognition rate on the two set of test images. Set 1 is consisted of 24 unseen patterns from Yale Face Database and Set 2 is consisted of 9,000 noisy images generated from the training images.

Table Recognition rate on two set of test images vs. three different methods

	W/ Write-verify	W/O Write-verify	Standard System
Set 1	22/24	21/24	22/24
Set 2	88.08%	85.04%	91.48%

Besides this, we also verified that the accuracy increased for all methods if we continued conducting the iteration.

Location of Change:

The abstract in the main text, from line 26 to line 29.

The introduction in the main text, from line 71 to line 74.

"Grey-scale face image classification" Section in the main text, from line 281 to line 282 and from line 287 to line 289.

“Analog RRAM array enables lower energy consumption” Section in the main text, from line 326 to line 328.

Comment 2:

I like that the authors provide the individual memristor response in the Supplementary materials (Fig. 3). Hopefully, the human synapses are less noisy.

Response:

Thanks for your comments. This is one direction which this device could be optimized.

Reviewer 3

We would like to thank the reviewer for the comments. According to the comments, we have made the revisions on the manuscript. Here is the point-by-point reply to the reviewer's Comments.

Comment 1:

This manuscript describes an experimental implementation of the on-chip training of a very small, single-layer perceptron neural network on a toy-size problem. The in-situ training is described, generalization and noisy inference is tested, and network accuracy is shown. The authors do share a large amount of data illustrating the high variability of their devices, which is good. Authors also focus overly much on the power advantages of such an online array, which of course are large -- yet do not provide useful information to other experts in the field, vis-a-vis: If we or they built a large array of these devices and used it to train a multi-layer neural network on a non-toy problem, would there be any chance of it actually working (e.g., converging to acceptably-interesting accuracies within any reasonable time)?

I recommend the authors go back and attempt to answer the question above using simulation of neural networks as informed by their extensive data on device-to-device variability and conductance response and provide the answer in their revised paper. Also, since I have very good reason to believe the answer to my question above is "No, it won't work with these devices," then authors will furthermore need to describe for the reader how much these devices need to be improved (in order to make non-toy DNN work), whether this is feasible, and how this could potentially be done.

Response:

We appreciate the questions and comments. The comments are really helpful to improve our work.

In this paper, we developed an analog 1T1R array which could perform stable bi-directional continuous conductance modulation. The optimized RRAM stack material is CMOS compatible which means the scale of the 1T1R array could increase significantly. The 1T1R cell structure enables reliable large-scale integration by suppressing the leakage current. This cell stacks and array architecture could be used in neuromorphic system as well as the hardware platform for deep learning. To display the array's potential, we demonstrated a perceptron network.

Our experimental demonstration consolidates the feasibility of analog synaptic array and accelerates the implementation of neuromorphic computing hardware system using new devices that may have advantage over convention digital memories. Yet there are still many issues to be solved before it realizes actual applications. Many groups have made great contributions by experimental and simulation studies. For example, IBM team have analyzed how non-ideal defects of PCM device affect the performance of the network (e.g. Reference 19 in the revision).

As suggested by the reviewer, we have simulated a more complex task. We have constructed a multi-layer-perceptron network with one single hidden layer to test on full MNIST database. This device model was extracted from the experimental data and has similar device-to-device, cycle-to-cycle variation and nonlinearity as the experimental devices. We found that with optimized algorithms, the learning is successful even though variability is large. We also agreed that suppressing variability and nonlinearity is important in the future work. The detail results are listed below according to the specific comments.

Comment 2:

The authors include the usual hand-waving rhetoric about the glories of neuromorphic computing, but of course low energy computation is completely useless if you are not obtaining a result of relevance. In this context, future neuromorphic systems built around such devices MUST be able to train the exact same size (or bigger) networks as are currently trained by GPUs, MUST obtain similar classification accuracies by the end of training, and most likely MUST provide a speedup advantage IN ADDITION TO lower power.

Yes, there is a niche market for solutions that train DNN at lower power to sufficient accuracy but more slowly than GPUs, but it is a fairly small niche. There is NO niche for solutions that fail to achieve similar classification accuracies on non-toy-size DNN. There is zero mention or acknowledgement of this reality in the current intro to this paper, and this must be fixed.

Response:

Thanks for pointing out this. We agree that the accuracy, speed and power are all important. We addressed this in the conclusion section and cited some related papers (Ref 38 in the revision).

We focus on implementation of large analog device array which could be integrated with CMOS circuits and used in neuromorphic system as well as hardware platform for deep learning. There are challenges in device optimization, architecture innovation, and algorithm development. At the same time, there are also significant upside potentials for drastically reducing the power consumption. As is well-known, computing speed today is essentially limited by power consumption. Integrating analog weight storage directly on top of computing elements significantly increases the bandwidth of memory access and eliminates power consumption of off-chip memory access^[1]. In this paper, we focused on the latency and energy cost of the 1024-1T1R array during the training process, including the inference phase and weight updating. The energy consumption of counterpart operations in a standard system is estimated to compare.

As for this recognition task, we achieved slightly lower recognition accuracy using standard CPU computing system. The convergence triggers as long as the perceptron recognizes all the training images. The following table shows the detail recognition rate on the two set of test images, indicating this experimental demonstration realized comparable recognition rate. Set 1 is consisted of 24 unseen patterns from Yale Face Database and Set 2 is consisted of 9,000

noisy images generated from the training images. Besides this, we verified that the accuracy increased for all methods if we continued conducting the iteration.

Table Recognition on two set of test images vs. three different methods

	W/ Write-verify	W/O Write-verify	Standard System
Set 1	22/24	21/24	22/24
Set 2	88.08%	85.04%	91.48%

The analog RRAM array has the natural speed and power advantages to do the inferencing. To achieve a comparable classification accuracy on larger network which are currently trained by GPUs and realize the superiority on power and speed at the same time, there are many technical issues to be solved. Reference 39 shows, using modeling, an analysis from system level and projects remarkable acceleration factors and power efficiency. This work illustrates, through experiments, the tradeoff among various device and algorithmic metrics such as device-to-device variability, cycle-to-cycle variability, the algorithms for RRAM weight update, and the design of the system architecture. And there lies the contribution of this work: an experimental evaluation and analysis of the tradeoff among various factors and identifying the key aspects that need to be researched on.

[1] Aly, M. M. S. et al. Energy-efficient abundant-data computing: The n3xt 1,000 x. Computer 48, 24-33 (2015).

Location of Change:

The abstract in the main text, from line 26 to line 29.

The introduction in the main text, from line 71 to line 74.

“Grey-scale face image classification” Section in the main text, from line 281 to line 282 and from line 287 to line 289.

“Analog RRAM array enables lower energy consumption” Section in the main text, from line 326 to line 328.

“Conclusion” Section in the main text, from line 346 to line 351.

Comment 3:

Authors claim "this is the first experimental demonstration of such a significant cognitive task." I get tired of papers that are explicitly built so that one can claim they are "the first," leaving all the actual hard work of taking something from first demonstration to usefulness to someone else. In this case, I cannot agree that the network shown here can be considered "significant."

This network and dataset is far too small to allow any predictive (or even indicative) power for larger non-toy networks. One can say that MNIST itself is also far smaller in size and easier in difficulty than CIFAR or ImageNet, but if a memristor-based solution were able to deliver identical-accuracy-performance on MNIST, I would have to claim that that accomplishment would be a major step towards the demonstration of NVM-based neuromorphic systems. The present network is far far smaller and much easier -- to the extent that success on this toy problem is indicative of nothing.

Response:

In this work, we focused on the implementation of bi-directional analog RRAM array which is CMOS compatible. This is the basis for future large-scale integration for a neuromorphic system. The network and dataset is small because of the limitation of the array size currently implemented in a device-technology experiment – a level of difficulty very different from putting together racks of GPUs. The energy cost comparison is a useful benchmark comparison that can be scaled up for larger systems since the energy consumption per iteration was compared.

Fig. R5. The Set and Reset performance using the device model. (a) The conductance increases gradually with respect to the identical Set pulses. (b) The conductance decreases gradually with respect to the identical Reset pulses.

A 784-300-10 multilayer perceptron (MLP) network was constructed to classify the MNIST patterns, as illustrated in Fig. R6. We extracted the device model from the test results. The simulated bidirectional conductance transferring performance is shown below. This model contains similar device-to-device variation, cycle-to-cycle variation, and nonlinearity with experimental data. The MNIST dataset contains of a training set of 60K and a test set of 10K different images. Each image is a 28×28 gray-scale pattern representing digits ranging from 0 to 9. ReLU function was adopted as the activation function.

Fig. R6. The schematic of the employed network model

The weights between input layer and hidden layer are implemented by a 784 x 300 device array, and the weights between hidden layer and output layer are implemented by another 300

x 10 device array. An elaborated BP algorithm and momentum term were used to optimize RRAM weights and further monolithic systems.

The conductance weights are updated using single SET pulse or RESET pulse (without write-verify scheme). The accuracy is 91.4%, a 4% decrease compared to the LeNet with the same architecture (Reference 20 in the revision). The nonlinearity of the device limited the increase of accuracy, as Reference 38 in the revision reported. The accuracy on the test set during training is presented below.

Fig. R7. The accuracy on test set during training using the without write-verify scheme and modified algorithm.

Location of Change:

The introduction in the main text, from line 50 to line 52.

Comment 4:

There is a major bait-and-switch between the aspect ratio of the experimental array (described as having 128 rows and 8 columns) and the logical array needed for the network of interest (described as having 3 rows and 320 columns). Thus statements like "A fully parallel read operation" is clearly a bald-faced lie, since there is absolutely no way the authors can experimentally be inputting all 320 columns at the same time given the dimensions of the physical array. Statements like this will need to be fixed to reflect the actual realities of the experiment as it was actually performed.

Response:

We operated the array row by row and this is what we meant by "a parallel read operation". We have changed the statement to avoid misunderstanding.

Location of Change:

"RRAM based neuromorphic network" Section in the main text, from line 108 to line 113 and line 144.

Comment 5:

Ways in which this experiment is a toy experiment- It is not that the images are too small - 20x20 is not far off in size from MNIST. The problem is that there are so few classes, which appear to be cherry-picked from the dataset in order to maximize the distance between the classes and make the problem easier.

Response:

In this work, the experiment was conducted to validate the developed bi-directional analog RRAM array is promising in running neural networks. And in the main text, we demonstrated gray-level human face pattern recognition using the Yale Face Database. Due to the limit of current array size, 3 classes from the database was recognized. We believe this demonstration is another important step on the basis of the previous work reported in “Nature 521, 61-64 (2015)” in which the black-and-white 3x3 letter images were recognized.

Comment 6:

Batch-based processing (e.g., going through the whole dataset and THEN updating all the weights) has long been abandoned by the AI community. IF you wanted to do this for a large dataset, you would take a large hit in convergence rate, plus you must describe how this data will be stored locally (or include the power for aggregating this data offline in your power and latency assessment).

Response:

Our work is focused on the analog weight storage array, the experimental demonstration of the array, and how the characteristics of the array may influence the efficacy of recognition. As such, the algorithm chosen to illustrate learning and recognition is not central to the discussion. Whether using batch method or not does not influence the key point of this work.

As such, the use of batch-based processing does not affect the energy consumption comparison either, as the related energy for such operations was not taken into consideration for the comparison.

Comment 7:

The convergence of this learning appears to be so rapid that the authors are able to almost completely avoid having to use any SET pulses during the experiment. This will most definitely not be the case for any non-toy experiment.

(Authors should show the histogram of desired conductance changes during training, so readers will know what the balance between requests-for-lower-conductances and requests-for-higher-conductances).

Response:

We checked the experiment data and found that 19.3% of the devices experienced SET transition during learning under the write-verify scheme. The weights evolution trace of ten randomly selected cells are shown in Fig. R8 (left two columns, in red), together with another

ten randomly selected cells (right two columns, in blue) from the rest cells which merely experienced RESET transitions.

Under the without write-verify scheme, the percentage of the devices which experienced SET during learning is 14.6%. The weights evolution trace of 20 randomly selected cells is presented in Fig.R9.

Fig. R8. Conductance evolution of 20 randomly selected RRAM devices during learning process under the write-verify scheme. The figures with red lines indicates the cells which experienced SET processes. And the figures with blue lines indicate the cells which merely experienced RESET processes.

Fig. R9. Conductance evolution of 20 randomly selected RRAM devices during learning process under the without write-verify scheme. The figures with red lines indicates the cells which experienced SET processes. And the figures with blue lines indicates the cells which merely experienced RESET processes.

Location of Change:

“Grey-scale face image classification” Section in the main text, from line 256 to line 257.
 Section 11 in Supplementary Information, from line 158 to line 164.

Comment 8:

Because the dataset is toy-size, authors probably terminated training too early, thus losing the opportunity for further improvements in generalization performance or classification of noisy inputs.

Response:

Thanks for pointing out this. It is exactly right that the generalization performance or classification of noisy inputs would be improved with more iterations. We have done a simulation for this network based on the device model under without write-verify scheme. This time the training stopped at 100th iteration instead of stopping at the epoch when it converged on the training set. The change of accuracy on training set, unseen test set (24 images) and noisy test set (9,000 images with up to 31.25% noisy pixels) with respected to iteration is shown below.

The training went converging on training set at 69th epoch. If we still went on the train until 100th iteration, the misrecognition rate on unseen test set and noisy pattern set decreased to 0% and 1.59% respectively.

Fig. R10. The evolution of accuracy with respected to iteration on training set, unseen test set and noisy test set.

Comment 9:

The hundreds of write-read-verify cycles required for these devices are not going to scale well for a non-toy problem, without some trick which allows you to do this extremely infrequently and/or efficiently.

Response:

We proposed two operating schemes, one with write-verify while the other without. The write-verify scheme could result in relatively precise conductance change but may be not

efficient. The without write-verify scheme is efficient in system design and easy to implement the peripheral circuits, but requires modification on current algorithms to be suitable for the device performance for actual applications. The two schemes might be suitable for different application scenario.

Besides, we also simulated MNIST database learning under without write-verify scheme, and get success with an improved algorithm as discussed above. This indicates that such analog RRAM array can work well on a relative complex task without any verification.

Comment 10:

If authors are going to focus on power to this extent, they must include the power required to implement their tanh() function, to compute the delta values after each example, and to aggregate and store the weight updates during their batch-based programming.

Response:

Thanks for pointing this. In our manuscript, we pay attention to the energy consumption of operations that are experimentally performed on RRAM array; namely, multiply-add operations (RRAM read) and weight update operations (outer products). And we only consider the energy of the same operations for a standard system; including both the processor operations and memory accesses needed for these operations for the standard system. To avoid confusion, in the revised manuscript, we describe this comparison in detail.

Location of Change:

Section 16 in Supplementary Information, from line 191 to line 198.

Comment 11:

The RRAM devices do seem to show a nice gentle behavior. This reviewer is impressed at the slick usage of a logarithmic horizontal axis to make the conductance response "look" linear, when in actuality it is quite far from linear. That said, the supplementary data in S4 S5 and S6 (showing just how bad the variability between these devices really is) is greatly appreciated by the reviewer. After seeing this data, I was amazed that even this toy experiment worked at all, until I realized that the toy database only required RESET pulses, and that in the verify version of your experiment, you were willing to wait as long as it took, even if that meant many hundreds of programming pulses.

Response:

The conductance changing with respected to identical pulse train is nonlinear. As for this demonstration, the devices did not merely require RESET pulses. More detail is shown in our reply to the early comment.

Both programming schemes led to successful convergence. The write-verify method employs identical pulse train to realize the quantized weight change. We configured a boundary for the pulse number which is shown in Fig. S9 in the Supplementary Information. During this demonstration, there were some devices that cannot reach the target weights and the system

still converged successfully. We did not need to wait for a long period. The pulse number boundary could be set to a smaller value to speed up the training. On the other hand, the pulse amplitude did affect the verify speed as shown in Fig 3F and 3G. A larger pulse amplitude could be adopted to accelerate the training and other effects on the system should be further studied.

Comment 12:

One comment on Figure 3B,C -- it is disconcerting that the vertical axes of B and C are not the same scale. You should either start your RESET characteristic at the same peak value that the SET characteristic terminates at, OR at least make the two plots have the same vertical scale so that the reader can immediately observe that the SET and RESET characteristics are NOT completing each other.

Response:

Since we want to keep the same horizontal axis range, we used different vertical axis ranges in Fig 3B and 3C. In the supplementary material, we have added the measurement using continuous SET and RESET pulse cycles and plotted them in a figure with the same vertical axis. The result is shown below.

Fig. R11. The conductance changes under Set and Reset pulse cycles.

Location of Change:

“Realization of bidirectional analog RRAM array” Section in the main text, from line 199 to line 200.

Section 6 in Supplementary Information, from line 104 to line 109.

Comment 13:

Assuming that there were even any SET pulses needed during the "training" of this toy problem, it would be good to show the "SET" versions of Figures 3F, G.

Response:

As we replied above, there are some cells requiring SET operation. And we conducted similar measurements for SET process and added the result to the supplementary material.

During the test, a sequence of identical SET pulses with write-verify were applied to examine how pulse amplitudes affect weight adjustment. The raw data was statistically averaged over 32 random chosen cells under 3 repeated procedures to get rid of device variances. The procedure started with precisely initializing cell conductance at $4 \mu\text{S}$ ($250 \text{ k}\Omega$). Then a specified pulse train was applied on bit line to tune cell conductance to a certain target value and recorded the refined conductance value and total pulse number when write-verify passed. The programming pulse width was 50 ns , the gate voltage V_{wl} was 2.8 V . The source line was grounded and the pulse number limitation was set to 300. Several trials were conducted during each test, tuning the 32 cells conductance to a same target conductance target set as the RESET test, i.e. $33.3 \mu\text{S}$ ($30 \text{ k}\Omega$), $28.6 \mu\text{S}$ ($35 \text{ k}\Omega$), $25 \mu\text{S}$ ($40 \text{ k}\Omega$), $22.2 \mu\text{S}$ ($45 \text{ k}\Omega$), $20 \mu\text{S}$ ($50 \text{ k}\Omega$), $18.2 \mu\text{S}$ ($55 \text{ k}\Omega$), $13.3 \mu\text{S}$ ($75 \text{ k}\Omega$) and $10 \mu\text{S}$ ($100 \text{ k}\Omega$) respectively. The results are shown below. We can conclude that there is a tradeoff between tuning speed, conductance modulation range and tuning accuracy when determining the SET pulse conditions.

Fig. R12. (A) The precision measurement result during SET process using verified pulse train with different amplitudes. (B) The number of pulses needed to reach the target conductance from the same initial state $4 \mu\text{S}$, showing the tuning speed with respect to different programming pulse amplitudes. (C) The conductance modulation range measurement during write-verify SET process under different pulse amplitudes.

Location of Change:

“Realization of bidirectional analog RRAM array” Section in the main text, from line 228 to line 230.

Section 10 in Supplementary Information, from line 139 to line 157.

Comment 14:

The results with classification of noisy training data and with the test data (non-training data) are appreciated, and help improve the quality of the paper. Two points here:

Since the 9000 noisy images are noise added to the training images, they should NOT be referred to EVER as "test patterns" (see line 269). It is VERY important that it be 100% clear this is noise on already-seen training images, NOT on never-before-seen test data.

To differentiate the statistically relevant percentage numbers in Figure 5b (out of 9,000 noisy images) from the generalization "percentages" in Figure 5c, please label the green bars as

generalization accuracy and include the exact fraction of correct generalized images (e.g., 22/24, etc.) as well as the percentage (to make it clear these "percentages" are accumulated over a very small number of instances).

Response:

Thanks very much for the comment. To make a difference, we will call the noisy patterns generated from the training patterns “augmented noisy test patterns”, and call the other “unseen test patterns”. Besides Figure 5c is modified as below.

Fig. R13. The comparison between with (W/) and without (W/O) schemes in terms of latency, energy consumption during training process and test accuracy on unseen test patterns in normalized format.

Location of Change:

“Grey-scale face image classification” Section in the main text, line 284.
 Fig. 5C in the main text, line 307.

Comment 15:

Authors take great pains to laud the great power of experimental results, and similarly spend significant effort disparaging simulation-based efforts (including Ref 36 and the relevant portions of Ref 19, which also included a MUCH larger "first ever" experiment than the present manuscript).

However, this reviewer would claim that simulations -- especially ones that are done well enough that they can project accurately for the field whether the devices at hand will succeed or will fail at tasks of actual interest -- are much more useful for the field than experimental demonstrations at toy scale without such simulations. In most fields, toy experiments have little forward predictive power for tasks of actual interest -- for neural networks, they have zero forward predictive power.

Response:

We modified the corresponding part. We appreciate the simulation work including Ref. 19, Ref. 36 (cited as Ref. 39 in the revision), and so on. Ref. 19 analyzed how device defects affect the system and indicated how device should be improved, and Ref 36 showed the power and latency advantages and system design consideration from a high-level perspective. The

researches by simulation or experiment are all important to bring breakthrough in this field. Our experimental demonstration is assistant proof for the realization and application of such a bi-directional analog device array. It is capable of integrating to a large scale size with CMOS system which could be used in neuromorphic computing. There are still many issues to make this true and both simulation and experiment efforts should be appreciated.

Location of Change:

“Grey-scale face image classification” Section in the main text, from line 291 to line 292.

Comment 16:

That said, it would not take much for the authors of this manuscript to take their extensive experimental data and to project through simulation whether these devices as they stand (nonlinear conductance response AND massive device-to-device variability) would suffice for even easy non-toy problems like MNIST. Such simulations would also determine whether the nonlinear conductance response itself were invalidating --- e.g., if every single device in a large array functioned exactly like the device described by Figure 3B,C with ZERO variability, even in that ideal case would one have a shot at training MNIST successfully?

This is the most important and relevant question, which remains unanswered (or even acknowledged) by the present manuscript. This is why the manuscript should not be accepted as it currently stands. Without this kind of information, the reviewer cannot agree that "these experimental results pave the way" towards the desired energy-efficient neuromorphic systems.

Response:

The perceptron network converged successfully under two operating schemes by experiments. The MLP network with single hidden layer also achieved an acceptable accuracy of 91.4% by simulation using the nonlinear device model with large variability. However just as Reference 38 in the revision reported, the nonlinearity did limit the increase of accuracy and we should think more about the optimization of device, operation, algorithm and system architecture.

Location of Change:

The introduction in the main text, from line 72 to line 74.

“Conclusion” Section in the main text, from line 346 to line 351.

Reviewer 1

Comment:

The authors answered all the points raised by the reviewer and consider all the suggestions comments raised by the other reviewer. The paper is acceptable for publication in nature communication

Response:

We appreciate your positive comment. Thanks!

Reviewer 2

Comment:

The revised version of the manuscript takes into account all my suggestions and comments. Therefore, I believe that it is suitable for publication in Nature Communications.

Response:

Thanks for your comment.

Reviewer 3

Comment:

Authors responded to many of my points, and new data described that addressed a number of my complaints.

I can recommend this revise manuscript for acceptance.

Response:

Thanks for your positive comments.